# Explaining Graph Neural Networks for Node Similarity on Graphs

**Daniel Daza**[*]                                                                *d.dazacruz@vu.nl*
*Vrije Universiteit Amsterdam, The Netherlands*

**Cuong Xuan Chu**                                                      *cuongxuan.chu@de.bosch.com*
*Bosch Center for Artificial Intelligence, Germany*

**Trung-Kien Tran**                                                      *trungkien.tran@de.bosch.com*
*Bosch Center for Artificial Intelligence, Germany*

**Daria Stepanova**                                                     *daria.stepanova@de.bosch.com*
*Bosch Center for Artificial Intelligence, Germany*

**Michael Cochez**                                                             *michael.cochez@abo.fi*
*ELLIS Institute Finland & Abo Akademi University, Turku, Finland & Elsevier discovery lab, Amsterdam*

**Paul Groth**                                                                  *p.t.groth@uva.nl*
*University of Amsterdam, The Netherlands*

**Reviewed on OpenReview:** *https://openreview.net/forum?id=zDEwl4zidP*

## Abstract

Similarity search is a fundamental task for exploiting information in various applications dealing with graph data, such as citation networks or knowledge graphs. Prior work on the explainability of graph neural networks (GNNs) has focused on supervised tasks, such as node classification and link prediction. However, the challenge of explaining similarities between node embeddings has been left unaddressed. We take a step towards filling this gap by formulating the problem, identifying desirable properties of explanations of similarity, and proposing intervention-based metrics that qualitatively assess them. Using our framework, we evaluate the performance of representative methods for explaining GNNs, based on the concepts of mutual information (MI) and gradient-based (GB) explanations. We find that unlike MI explanations, GB explanations have three desirable properties. First, they are *actionable*: selecting particular inputs results in predictable changes in similarity scores of corresponding nodes. Second, they are *consistent*: the effect of selecting certain inputs hardly overlaps with the effect of discarding them. Third, they can be pruned significantly to obtain *sparse* explanations that retain the effect on similarity scores. These important findings highlight the utility of our metrics as a framework for evaluating the quality of explanations of node similarities in GNNs.

Our implementation is available at https://github.com/dfdazac/exigraph.

## 1 Introduction

Graphs provide a powerful and expressive data structure for modeling relations between objects across diverse domains, such as social networks, biological systems, and knowledge bases (Newman, 2018; Hamilton et al., 2017a; Nickel et al., 2016; Hogan et al., 2021). Additionally, their ability to represent entities and their

---

[*]Work done during internship at the Bosch Center for AI.

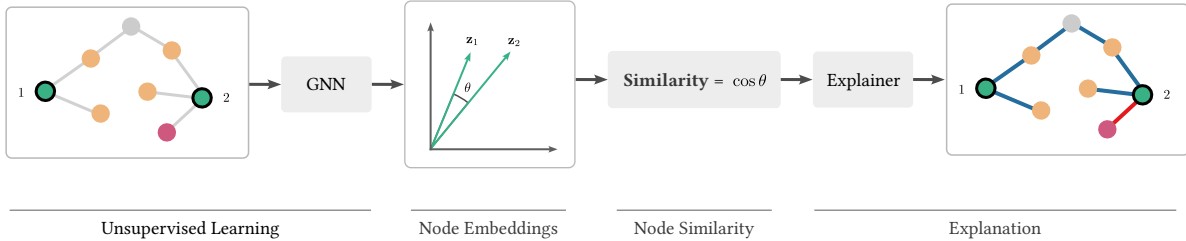

Figure 1: Illustration of the problem we investigate in our work. Given nodes 1 and 2 in a graph, unsupervised learning methods can be used to train a GNN to learn node embeddings, where a score of similarity can be estimated by cosine similarity. We investigate how to create explanations for such scores, that assign values of attributions to edges in the graph. In this example, a positive influence on the similarity score is shown in blue, and a negative influence is shown in red.

interactions makes them a natural representation for machine learning methods that seek to learn structural patterns and use them in downstream tasks.

A fundamental problem that arises across domains is similarity search, where the goal is to identify objects that resemble a given query object according to structural or semantic notions of similarity. In particular, we are concerned with similarity search over graphs, where given a query node, the goal is to retrieve a ranked list of similar nodes. Several methods to solve this problem have been proposed in the literature, ranging from heuristic-based methods to data-driven machine learning methods. Heuristics for similarity search on graphs exploit various graph statistics or techniques based on hashing to solve the problem (Shimomura et al., 2021; Shi et al., 2021).

Machine learning methods, on the other hand, avert the need to design handcrafted heuristics or features. Instead, they seek to exploit domain-specific patterns in the graph to learn node representations, or *embeddings*, after which similarities are captured via metrics such as cosine similarity on these representations. Graph neural networks (GNNs), in particular, have become a standard in machine learning approaches that process graph-structured data (Kipf & Welling, 2017; Schlichtkrull et al., 2018; Gilmer et al., 2017).

While GNNs offer several advantages due to their capacity to adapt to specific properties of the graph at hand, these benefits may be compromised when interpretability becomes a necessity (Burkart & Huber, 2021; Arrieta et al., 2020). Given their demonstrated effectiveness on different tasks, there are compelling motivations to explore methods for explaining their predictions (Yuan et al., 2023), which would enable applications that require accountable decision-making to leverage their predictive power.

While extensive works on explaining GNNs exist, the majority of the methods focus on supervised learning problems, where the predicted target is well-defined based on some ground-truth data, as in the case of node classification (Ying et al., 2019; Luo et al., 2020; Lucic et al., 2022; Miao et al., 2022). To the best of our knowledge, the applicability of such methods to the problem of explaining node similarities remains an open question.

Fig. 1 illustrates this problem, where a learning algorithm is used to train a GNN for computing embeddings for nodes 1 and 2. The embeddings are used to compute the cosine similarity that we aim at explaining. The explanation consists of an attribution of values to edges, depending on their influence on the similarity score. In the example, blue edges result in increasing similarity scores and red edges result in decreasing the score. Depending on the explanation method used, the effect of attribution values on similarity scores can be different.

Our work takes a step towards understanding what it means to *explain node similarity* in graph neural networks. This setting differs fundamentally from the supervised tasks for which existing GNN explainers were designed (Yuan et al., 2023). While prior work focuses on discrete predictions, such as node classification or link prediction, explaining *continuous similarity scores* is a fundamentally different problem that has not been addressed by current evaluation protocols. We bridge this gap and provide the following contributions:

- We introduce and study the problem of explaining node similarity in GNNs, which to the best of our knowledge, has not been addressed in prior work.

- Building on principles from explainable AI, we derive three criteria for explanations of node similarity. We then propose model-agnostic metrics that quantify these criteria by measuring how explanations behave under controlled interventions on the graph.

- We demonstrate the utility of our framework by applying it to representative mutual-information and gradient-based methods, yielding several important insights that demonstrate the practical benefits of our criteria and metrics.

## 2 Related work

**Similarity learning.** The problem of computing node similarities on graphs has been addressed in previous methods that rely on heuristics, rather than representations learned from the data. Some examples of such methods rely on statistics of connectivity (Brin, 1998; Haveliwala, 2002), co-occurrence statistics (Jeh & Widom, 2002), meta-paths in heterogeneous networks (Sun et al., 2011), and metrics for measuring structural similarities (Xu et al., 2007). Other methods employ ideas from hashing techniques to compute vector representations useful for similarity search (Gionis et al., 1999; Zadeh & Goel, 2013; Shimomura et al., 2021). Such heuristics are beneficial when they are broad enough to be applicable to different graphs. Graph neural networks, on the other hand, are able to adapt to specific signals present in the data, such as domain-specific topological properties and rich multi-modal features like text and images (Markowitz et al., 2022; Gao et al., 2020). Their demonstrated effectiveness for different tasks thus warrants an investigation on how explanations can be provided for them, in the event of applications where rationales for predictions of GNNs are valuable, which is the open question we address in this work.

**Unsupervised learning on graphs.** In contrast to tasks like node classification or regression where labeled data is available, similarity learning is rarely accompanied by ground truth data. An alternative is concerned with learning representations that capture patterns already present in the graph (Liu et al., 2023a; Xie et al., 2023; Liu et al., 2023b). In the absence of labels that could be used for training, learning in this setting relies on optimization algorithms that produce representations useful for a pretext task. Examples of pretext tasks are maximizing the mutual information between different views of a graph (Velickovic et al., 2019; Sun et al., 2020; Peng et al., 2020), embedding shortest path distances (Bojchevski & Günnemann, 2018; Frogner et al., 2019), reconstructing parts of the input (Kipf & Welling, 2016; Wang et al., 2017a), or maintaining invariance with respect to small changes in the input (Thakoor et al., 2022; Xie et al., 2022b). The resulting representations can then be employed in tasks such as clustering and similarity search.

Most of the research in this area has focused on studying different ways of designing pretext tasks. However, the area of explainability in unsupervised learning on graphs is underexplored (Xie et al., 2023; Liu et al., 2023b). A recently proposed method is Task-Agnostic Graph Explanations (TAGE) (Xie et al., 2022a), which proposes explaining specific dimensions of embeddings obtained via unsupervised learning. The motivation for explaining embedding dimensions is transferring the explainer module of TAGE to supervised learning tasks. The performance of TAGE for generating explanations for problems where labeled data is not available, such as similarity computations, has been so far left unexplored.

In our work, we focus on evaluating explanations of similarity in the unsupervised learning setting, which is the problem that has not been explored in TAGE or any prior work on unsupervised learning on graphs, and has also been acknowledged in comprehensive reviews in this area (Liu et al., 2023b; Xie et al., 2023).

**Explaining graph neural networks.** Graph neural networks (GNNs) are neural networks tailored to the irregular structure of graphs, that are able to learn representations of a node in a graph taking into consideration arbitrary subgraphs around it (Zhou et al., 2020; Wu et al., 2021; Ye et al., 2022). A growing number of methods have been proposed in the literature that provide explanations to predictions computed by GNNs, in the form of edges and features responsible for a prediction (Yuan et al., 2023). Existing methods assume a trained GNN and provide *post hoc* mechanisms for explaining their predictions (Ying et al., 2019;

Luo et al., 2020; Duval & Malliaros, 2021; Yuan et al., 2021; Muschalik et al., 2025), or propose methods that are explainable a priori (Miao et al., 2022; Lee et al., 2023). Fundamentally, these methods focus on devising mechanisms for explaining supervised tasks, such as node or graph classification. In our work, we instead focus on how this problem differs from the task of node similarity, and how to evaluate such methods for the respective task.

Orthogonally, approaches for *data valuation* have proposed methods for identifying how "valuable" certain parts of a graph (e.g., nodes or edges) are for a GNN, which can be defined in terms of changes to its learned weights (Chen et al., 2023); or its performance for node classification (Song et al., 2023; Chi et al., 2025). Our goal is complementary to these works: rather than studying the effect of graph components on GNN parameter updates or classification performance, we analyze how they influence the computation of individual similarity scores. This constitutes a fundamentally different, label-free setting, where similarity is a continuous quantity and standard notions of prediction correctness or task utility are not as well defined as in accuracy metrics for node classification.

More recently, Piaggesi et al. (2025) proposed metrics for evaluating the interpretability of unsupervised node embeddings. These metrics are designed to assess the semantic structure of the latent space produced by a specific representation learning method, and require ground-truth graph annotations (e.g., communities or motifs). In contrast, our objective is not to evaluate embeddings but rather to evaluate explanations of **similarity scores** produced by arbitrary GNNs. Our metrics are model-agnostic and operate by measuring the effect of graph interventions; therefore, they quantify the quality of explanations when defined over the structure of the graph rather than the embedding space. As such, the two sets of metrics address complementary but fundamentally different questions.

**Knowledge graph embeddings and entity similarity.** Knowledge graph embeddings are representations of entities and relation types, which are commonly trained for the *link prediction* task (Nickel et al., 2016; Wang et al., 2017b): Given a query entity and a relation, the embeddings are used to predict a target entity that is likely to form a valid triple with the query entity and relation. KG embeddings have been applied in similarity computations via functions like cosine similarity or the dot product (Liu et al., 2019; Yamada et al., 2020; Gerritse et al., 2020; Daza et al., 2021; Khan et al., 2022), which are not designed to be explainable.

Prior work has explored the problem of explainability for KG embeddings. Some methods have proposed learning embeddings with a predefined structure, such as a set of interpretable concepts (Chandrahas et al., 2020; Xie et al., 2017; Zhang et al., 2021), or via sparsity constraints (Zulaika et al., 2022). The result is an embedding space, where it is possible to identify distinct semantic regions, e.g., "professions" or "cities". This differs from the problem of grounding similarities computed between pairs of entities on known attributes of the entities, which is the focus of our work.

In several other works, given an existing set of KG embeddings trained for link prediction, explanations have taken the form of a subset of supporting triples (Zhang et al., 2019; Pezeshkpour et al., 2019; Betz et al., 2022; Rossi et al., 2022), paths (Gusmão et al., 2018), or Horn rules (Gad-Elrab et al., 2020). While there is empirical evidence for KG embeddings being able to capture notions of similarity (Gad-Elrab et al., 2020), some works have suggested that the link prediction objective is sub-optimal for this task (Ristoski & Paulheim, 2016; Cochez et al., 2017; Ristoski et al., 2019). This motivates our use of GNNs that operate directly on node features and subgraphs, which can serve as explanations for predicted similarity scores.

Another line of work (Petrova et al., 2017; 2019) focused on identifying the reasons behind the similarity of two given entities by extracting SPARQL queries, which have both entities as answers. However, unlike in our proposal, in (Petrova et al., 2017; 2019) the authors did not aim at explaining the similarity scores computed by a machine learning method, but rather exclusively relied on the graph structure.

## 3 Learning and explaining similarities

Let $G = (\mathbf{A}, \mathbf{X})$ be a graph with $n$ nodes, where $\mathbf{A}$ is an $n \times n$ adjacency matrix with $A_{ij} = 1$ if nodes $i$ and $j$ are connected, and 0 otherwise, and $\mathbf{X} \in \mathbb{R}^{n \times m}$ is a feature matrix, where the $i$-th row $\mathbf{x}_i$ contains the

$m$-dimensional feature vector of the node $i$. In the following sections, we discuss the problems of learning representations of nodes for the similarity task, and our proposals on how similarity scores can be explained.

## 3.1 Learning representations for similarity

Graph neural networks have become a standard architecture for processing graph-structured data, due to their ability to incorporate arbitrary neighborhoods around a node (Kipf & Welling, 2017; Gilmer et al., 2017; Xu et al., 2019; Maron et al., 2019; Corso et al., 2020). They can easily be extended to graphs with rich edge features and multimodal data (Schlichtkrull et al., 2018; Saqur & Narasimhan, 2020; Galkin et al., 2020; Ektefaie et al., 2023). Furthermore, the fact that GNNs implement an explicit function that maps node neighborhoods and features to an embedding offers the opportunity for determining which parts of the input are responsible for a certain output. This is a desirable property when explaining computations such as similarity scores.

A prominent example of a graph neural network is the Graph Convolutional Network (GCN) (Kipf & Welling, 2017). A single layer of the GCN implements the following propagation rule:

$$\text{GCN}(\mathbf{X}, \mathbf{A}) = \sigma\left(\tilde{\mathbf{A}}\mathbf{X}\boldsymbol{\Theta}\right), \tag{1}$$

where $\tilde{\mathbf{A}}$ is the normalized adjacency matrix, $\tilde{\mathbf{A}} = \hat{\mathbf{D}}^{-\frac{1}{2}}\hat{\mathbf{A}}\hat{\mathbf{D}}^{-\frac{1}{2}}$. Let $\mathbf{I}_n$ be the $n \times n$ identity matrix. Then $\hat{\mathbf{A}} = \mathbf{A} + \mathbf{I}_n$ is the adjacency matrix, adding self-loops, and $\hat{\mathbf{D}}$ is the degree matrix after adding self loops, such that $\hat{D}_{ii} = \sum_j \hat{A}_{ij}$.

The weight matrix $\boldsymbol{\Theta}$ in Eq. 1 contains the parameters of the layer to be learned during training. When composing together multiple GCN layers, we obtain a function $f_\theta(\mathbf{X}, \mathbf{A}) = \mathbf{Z} \in \mathbb{R}^{n \times d}$ that maps each node and its features to an embedding, conditioned on the features of nodes in its neighborhood.

We approach the problem of training a GNN to learn node embeddings from the perspective of unsupervised learning: In the absence of labeled data containing ground-truth similarity information, we resort to methods that learn node embeddings by capturing patterns existing in the graph, such as communities or structural roles (Hamilton et al., 2017a). The resulting node embeddings are vectors $\mathbf{z}_i \in \mathbb{R}^d$, with $i = 1, \dots n$, where such patterns are preserved by the geometry of the space. This allows us to address the problem of similarity search for a given query node $i$, by ranking the rest of the nodes in the graph according to a function such as cosine similarity:

$$y(i, j) = \frac{\mathbf{z}_i^\top \mathbf{z}_j}{\|\mathbf{z}_i\|\|\mathbf{z}_j\|}, \tag{2}$$

where $j = 1, \dots, n$ and $\|\mathbf{z}_i\|$ is the $\ell^2$-norm of $\mathbf{z}_i$.

Several methods are available in the literature for unsupervised learning on graphs (Hamilton et al., 2017a; Liu et al., 2023b; Ju et al., 2023). Examples include Graph Autoencoders and Variational Graph Autoencoders (Kipf & Welling, 2016), which optimize node embeddings so that they are able to reconstruct the adjacency matrix; Deep Graph Infomax (Velickovic et al., 2019), that learns node embeddings by maximizing the mutual information between them and a summarized representation of the graph; and Graph Contrastive Representation Learning (Zhu et al., 2020), which compares different views of a node by perturbing its neighborhood and features.

## 3.2 Explaining GNNs

The success of GNNs at various tasks has been accompanied by increased interest in explaining the predictions they provide (Yuan et al., 2023). Informally, methods for explaining GNNs aim to determine i) which parts of the input graph $G = (\mathbf{X}, \mathbf{A})$ are responsible for a particular prediction, and ii) how they are responsible. The mechanisms used to answer these questions vary with each method.

A recent survey (Yuan et al., 2023) classifies methods for explaining GNNs into two main groups: instance-level and model-level methods. Instance-level methods produce a distinct explanation for a particular pre-

diction (such as the label predicted for a specific node in the graph), while model-level methods aim to understand the behavior of the GNN under different inputs. Since we are interested in explaining similarity scores computed for specific pairs of nodes, we focus on the class of instance-level explanations.

Examples of instance-level methods are perturbation methods and gradient-based methods (Yuan et al., 2023). They represent an explanation as an assignment of values to parts of the input (for example, edges in the graph or node features), where the values indicate a degree of importance for computing the output of the GNN, as we illustrate in Fig. 1. In this work, the parts of the inputs to the GNN that we consider for explanations are edges between nodes, but our discussion can be easily extended to consider node features.

Formally, we assume that we have access to an already trained GNN. The output $f_\theta(\mathbf{X}, \mathbf{A})$ of the GNN is used to compute a *prediction* $y = g(f_\theta(\mathbf{X}, \mathbf{A}))$, and we wish to compute an explanation for it that describes the degree of influence of an edge in a prediction. For similarity search the prediction is the cosine similarity between two specific node embeddings as defined in Eq. 2.

Explanations over edges in the graph can be defined as a function that maps a prediction to a matrix $\mathbf{M} \in \mathbb{R}^{n \times n}$ containing *explanation values* for each of the (non-zero) entries of the adjacency matrix. For the majority of perturbation methods, the explanation values in $\mathbf{M}$ lie in the interval $[0, 1]$, and they can be interpreted as a *mask*, where values of 1 indicate relevant edges and 0 irrelevant ones. Gradient-based methods, on the other hand, are unconstrained, providing explanation values over the real numbers that not only carry the magnitude with which an edge influences a prediction, but also its direction (positive or negative) via the sign of the gradient.

Given a matrix $\mathbf{M}$ of explanation values, a subset of the edges in the graph can be selected by defining an *explanation threshold* $t$. The subset is defined by the entries in the adjacency matrix $A_{ij}$ such that $M_{ij} > t$. The meaning of the selected edges for an explanation of the similarity score depends on whether the matrix is interpreted as a mask, or as a gradient.

**Explaining node similarities.**   Prior work on explaining GNNs has primarily focused on supervised tasks like node classification and link prediction (Yuan et al., 2023), where the goal is to justify a discrete output label. In contrast, node similarity is a continuous, unsupervised quantity, and existing GNN explainers do not come with criteria which a good similarity explanation should satisfy. To address this gap, we revisit principles from the general explainable AI literature and adapt them to the problem of explaining similarity scores. We identify a set of criteria that such explanations should meet and introduce metrics that allow us to quantitatively evaluate them.

## 4   Criteria for explanations of similarity

Several works in the literature have highlighted the importance of explainability in artificial intelligence systems, particularly when they face human users that could benefit from an understanding of their predictions (Ras et al., 2018; Mueller et al., 2019; Miller, 2019; Arrieta et al., 2020; Yang et al., 2023). These works define a series of properties that explanations should have. For example, they should *"produce details or reasons to make its functioning clear or easy to understand"* (Arrieta et al., 2020), they should be useful for debugging algorithms (Yang et al., 2023), they should provide answers to *why* questions (Miller, 2019) (e.g., *why is this the similarity score?*); and they should have properties such as fidelity (i.e., how much the explanation agrees with the input-output map of the prediction under explanation), low ambiguity, and low complexity, among others  (Ras et al., 2018). In the context of node similarity in GNNs, several explanation methods assign a relevance value to each edge involved in the computation. We derive three criteria that such explanations should meet:

**1.   Actionable explanations.**   We can use the edges whose explanation value is above or below the threshold $t$ to make interventions in the graph that result in a predictable effect on the original similarity score. This facilitates an understanding of the specific effect of some edges on the similarity score, and follows requirements on understanding model decisions (Miller, 2019; Arrieta et al., 2020), interactivity via interventions (Arrieta et al., 2020), model debugging (Yang et al., 2023), and fidelity (Ras et al., 2018).

**2. Consistent explanations.** Actionability alone does not guarantee that the two sides of the threshold capture *distinct* explanatory behavior. An explanation can be actionable yet non-discriminative: both edges above and below a threshold $t$ may produce the same effect (e.g., both increase similarity). Thus, explanations should be consistent: the effect of keeping edges above the threshold is distinct from the effect of discarding them. This implies that the explanations capture specific behaviors of the similarity under explanation, indicating fidelity and low ambiguity (Ras et al., 2018).

**3. Sparse explanations.** Explanations should admit a principled reduction to a small subset, e.g., selecting the smallest subset that preserves 90% of the effect attributed to the intervention. This does not follow from actionability or consistency, as an explanation may satisfy both yet distribute its effect uniformly across many edges, making it impossible to reduce. Sparsity, therefore, captures the *compressibility* of an explanation while maintaining the effects that define its actionability and consistency. This leads to simpler, parsimonious explanations (Ras et al., 2018) that users can interpret (Miller, 2019).

We now introduce concrete, intervention-based metrics that operationalize these criteria and allow us to evaluate explanation methods quantitatively.

## 4.1 Metrics for intervention-based evaluation

Given a trained GNN $f_\theta$, we evaluate the properties of explanations for node similarities by measuring quantities that assess changes in the similarity score, after performing interventions in the graph on the basis of the explanation. More concretely, let $(i, j)$ be a pair of nodes in the graph. Given the set of node embeddings $\mathbf{Z} = f_\theta(\mathbf{X}, \mathbf{A})$, we select the embeddings of $i$ and $j$ from it and compute the cosine similarity $y(i, j)$ as defined in Eq. 2. The explanation method is then executed on this value, which results in an explanation matrix $\mathbf{M}$.

Given $\mathbf{M}$, we compute two matrices $\mathbf{M}_a$ and $\mathbf{M}_b$ that select values above or below a threshold $t$, respectively, such that

$$M_{a,ij} = M_{ij} \quad \text{if } M_{ij} \geq t \text{ else } 0 \tag{3}$$
$$M_{b,ij} = M_{ij} \quad \text{if } M_{ij} < t \text{ else } 0, \tag{4}$$

where the threshold for GNNexplainer is 0.5 and 0 for Gradient Based (GB) methods.

We use these matrices to intervene in the graph, by computing the element-wise multiplication of these matrices with the adjacency matrix, and re-computing the node embeddings, which yields

$$\mathbf{Z}_a = f_\theta(\mathbf{X}, \mathbf{A} \odot \mathbf{M}_a) \tag{5}$$
$$\mathbf{Z}_b = f_\theta(\mathbf{X}, \mathbf{A} \odot \mathbf{M}_b). \tag{6}$$

Given these embeddings, we then re-compute the similarity scores, which for each case we denote as $y_a(i, j)$ and $y_b(i, j)$ respectively.

**Measuring actionability.** Based on these new similarity scores, we first compute a *fidelity* metric (Ribeiro et al., 2016), which measures the change in the similarity score after the intervention with respect to the original similarity score:

$$\text{Fid}_a = y_a(i, j) - y(i, j) \tag{7}$$
$$\text{Fid}_b = y_b(i, j) - y(i, j) \tag{8}$$

Positive values of $\text{Fid}_a$ and $\text{Fid}_b$ indicate an increase in similarity, negative values indicate a decrease. If either intervention produces effects in a predictable direction, the explanation is **actionable**. Importantly, $\text{Fid}_a$ and $\text{Fid}_b$ capture absolute, independent changes, and they do not reveal how the two interventions relate.

**Measuring consistency.** To evaluate whether explanations induce distinct effects above vs. below the threshold, we consider only the signs of the fidelity values. For each node pair, we count $a_1$: times where $\text{Fid}_a$ is positive; $a_2$: times where $\text{Fid}_a$ is negative; and $b_1$ and $b_2$ defined similarly for $\text{Fid}_b$. We then define the Effect Overlap (EO) as the generalized Jaccard similarity between these counts:

$$\text{EO} = \frac{\sum_{i=1}^{2} \min(a_i, b_i)}{\sum_{i=1}^{2} \max(a_i, b_i)}. \tag{9}$$

An explanation method with an EO of zero indicates that the effect observed in $\text{Fid}_a$ is always positive, and always negative in $\text{Fid}_b$ (or vice versa). This indicates that the effects are distinct and thus the explanations are consistent. The maximum value of EO is 1 and it occurs if the effect is always positive or always negative, leading to an undistinguishable effect. Values between 0 and 1 indicate partial overlap.

EO thus resolves a limitation of fidelity: even if both $\text{Fid}_a$ and $\text{Fid}_b$ suggest actionability, EO determines whether the two interventions imply complementary effects on node similarity.

**Measuring sparsity.** Sparsity evaluates whether an explanation can be reduced to a smaller subset of edges while preserving the intervention effects that determine actionability and consistency. Given the explanation matrix $\mathbf{M}$ and the thresholded masks $\mathbf{M}_a$ and $\mathbf{M}_b$, we simulate different sparsity levels by removing a fraction $s \in [0, 1]$ of the least relevant edges: in $\mathbf{M}_a$ we drop the smallest $s$-fraction of nonzero values, and in $\mathbf{M}_b$ we drop the largest $s$-fraction. We then recompute the same fidelity and effect-overlap metrics across increasing values of $s$.

An explanation satisfies the sparsity criterion if its actionable and consistent behaviors are preserved as $s$ increases, indicating that the explanatory signal can be concentrated in a compact subset of edges.

## 5 Experiments

In our experiments, we aim to evaluate the proposed criteria and metrics in practice. To demonstrate this, we apply our framework to representative explanation methods from two major families of methods in the related work: mutual-information methods and gradient-based methods.

### 5.1 Explainability methods

#### 5.1.1 Mutual information methods

A common approach for identifying explanations for GNNs consists of determining what edges are relevant for computing a prediction, by relying on the concept of Mutual Information (MI)(Ying et al., 2019; Luo et al., 2020; Wang et al., 2021; Miao et al., 2022). Existing works have proposed explaining a prediction $y = g(f_\theta(\mathbf{X}, \mathbf{A}))$ by finding a subgraph from the original graph that has high mutual information with the prediction. This implies that only a region of the graph is relevant for computing a prediction, whereas the rest can be discarded without affecting it. This mechanism for finding an explanation can be formalized by assuming that the matrix $\mathbf{M}$ of explanation values is a sample of a random variable $M$ with values in $\{0, 1\}$, and then maximizing the mutual information between the original prediction (now a random variable $Y$) and the prediction after "masking" the adjacency matrix with the values in $M$:

$$\max_{M} I(g(f_\theta(\mathbf{X}, \mathbf{A})), g(f_\theta(\mathbf{X}, \mathbf{A} \odot M))) \tag{10}$$

where $\odot$ indicates element-wise multiplication.

In practice, the problem in Eq. 10 is not tractable. Instead, an approximation leads to the problem of finding a matrix that minimizes the cross-entropy loss (Ying et al., 2019):

$$\mathbf{M}_{\text{MI}} := \arg\min_{\mathbf{M}} -\mathbb{E}_Y[\log p(Y|\mathbf{X}, \mathbf{A} \odot \mathbf{M})] \tag{11}$$

This problem is solved by randomly initializing $\mathbf{M}_{\text{MI}}$ and updating it via gradient descent in the direction that minimizes the cross-entropy loss (Ying et al., 2019; Luo et al., 2020; Miao et al., 2022).

**Interpreting the explanation matrix.** Given the formulation of MI-based methods for explaining GNNs, entries of $\mathbf{M}_{\mathrm{MI}}$ with a value of 1 indicate edges that are relevant for the prediction, and 0 if they are irrelevant. When the matrix contains values in the continuous interval $[0, 1]$, an appropriate threshold for selecting or discarding edges is then $t = 0.5$.

In our experiments, we employ GNNExplainer (Ying et al., 2019) as an instance of MI methods.

### 5.1.2 Gradient-based methods

An early approach for identifying parts of the inputs relevant for a prediction computed by a neural network is to compute the gradient of the output with respect to the input (Simonyan et al., 2014; Shrikumar et al., 2017; Selvaraju et al., 2017; Sundararajan et al., 2017). This is motivated by the fact that the gradient indicates the direction and rate with which the outputs change with respect to the inputs.

In gradient-based (GB) methods, the extension of this approach to explaining GNNs is natural: the explanation matrix is equal to the gradient of the prediction with respect to the adjacency matrix,

$$\mathbf{M}_{\mathrm{GB}} \coloneqq \nabla_{\mathbf{A}} g(f_\theta(\mathbf{X}, \mathbf{A})). \tag{12}$$

Relying on the gradient alone might become problematic in deep neural networks using non-linearities like the ReLU activation function, whose derivative is zero over half of its domain. To address this issue, more advanced methods based on the gradient have been proposed, such as Guided Backpropagation (Springenberg et al., 2015), which ignores zero gradients, or Integrated Gradients (Sundararajan et al., 2017), which computes the total change from different values of the gradient, rather than relying on a single gradient.

**Interpreting the explanation matrix.** The values in the explanation matrix $\mathbf{M}_{\mathrm{GB}}$ are unconstrained, and they can take positive or negative values, depending on the sign of the gradient. This means that for each edge in the graph, GB explanations provide a magnitude and direction of influence. In this case, an appropriate threshold for selecting or discarding edges is $t = 0$.

When explaining predictions of node similarity, the $(i, j)$ entry of the explanation matrix indicates i) how much the presence of an edge between nodes $i$ and $j$ influences the similarity score, via the magnitude of the gradient, and ii) the direction of influence –positive or negative– via the sign. Unlike explanations from MI methods, we note that GB explanations are therefore more fine-grained, by providing additional information about how inputs affect changes in similarity scores.

In our experiments with GB methods, we consider direct gradient computation with respect to the adjacency matrix (as defined in Eq. 12), and Integrated Gradients (Sundararajan et al., 2017).

### 5.2 Node embedding methods

We implement the following unsupervised methods for learning node embeddings: Graph Autoencoders (GAE) and Variational Graph Autoencoders (VGAE) (Kipf & Welling, 2016), Deep Graph Infomax (DGI) (Velickovic et al., 2019), and Graph Contrastive Representation Learning (GRACE) (Zhu et al., 2020). We use them to train a 2-layer GCN (Kipf & Welling, 2017) as defined in Eq. 1. We tune hyperparameters via grid search, selecting the values with the lowest training loss.

### 5.3 Datasets

We run experiments with six graph datasets of different sizes and domains: Cora, Citeseer, and Pubmed (Sen et al., 2008; Namata et al., 2012; Yang et al., 2016) are citation networks from the computer science and medical domains, where each node corresponds to a scientific publication and an edge indicates that there is a citation from one publication to another. These graphs are known to exhibit high *homophily*: similar nodes (such as publications within the same field) are very likely to be connected (McPherson et al., 2001).

To consider graphs with different structural properties, we also carry out experiments with *heterophilic* graphs where connected nodes are not necessarily similar. Chameleon and Squirrel are graphs obtained from Wikipedia, where each node is a web page and an edge denotes a hyperlink between pages (Rozemberczki

Table 1: Statistics of graphs used in our experiments.

| Dataset | Nodes | Edges | Features |
|---|---|---|---|
| Cora | 2,708 | 5,429 | 1,433 |
| Citeseer | 3,327 | 4,732 | 3,703 |
| Pubmed | 19,717 | 44,338 | 500 |
| Chameleon | 2,277 | 36,101 | 2,325 |
| Actor | 7,600 | 33,544 | 931 |
| Squirrel | 5,201 | 217,073 | 2,089 |
| DBpedia50k | 30,449 | 57,161 | N/A |

Table 2: Results of fidelity metrics ($\text{Fid}_a$ and $\text{Fid}_b$) and effect overlap (EO, lower is better) when applying different explanation methods to multiple unsupervised learning methods and graphs. As explanation methods we consider GNNExplainer (Ying et al., 2019) (MI), and two gradient-based methods based on direct computation of the gradient (GB1), and Integrated Gradients (Sundararajan et al., 2017) (GB2).

| Method | | Cora $\text{Fid}_a$ | $\text{Fid}_b$ | EO | Citeseer $\text{Fid}_a$ | $\text{Fid}_b$ | EO | Pubmed $\text{Fid}_a$ | $\text{Fid}_b$ | EO | Chameleon $\text{Fid}_a$ | $\text{Fid}_b$ | EO | Actor $\text{Fid}_a$ | $\text{Fid}_b$ | EO | Squirrel $\text{Fid}_a$ | $\text{Fid}_b$ | EO |
|---|---|---|---|---|---|---|---|---|---|---|---|---|---|---|---|---|---|---|---|
| | MI | 0.133 | 0.019 | 0.451 | 0.130 | 0.029 | 0.406 | 0.136 | 0.202 | 0.532 | 0.292 | 0.353 | 0.531 | 0.134 | 0.209 | 0.521 | 0.386 | 0.357 | 0.411 |
| GAE | GB1 | 0.118 | -0.076 | 0.033 | 0.114 | -0.026 | 0.129 | 0.236 | -0.064 | 0.141 | 0.355 | -0.107 | 0.125 | 0.442 | -0.146 | 0.120 | 0.520 | -0.126 | 0.160 |
| | GB2 | 0.279 | -0.067 | **0.013** | 0.366 | -0.025 | **0.098** | 0.443 | -0.144 | **0.011** | 0.718 | -0.180 | **0.030** | 0.555 | -0.392 | **0.008** | 0.755 | -0.317 | **0.038** |
| | MI | 0.103 | 0.039 | 0.504 | 0.156 | 0.004 | 0.397 | 0.140 | 0.149 | 0.502 | 0.311 | 0.403 | 0.540 | 0.142 | 0.176 | 0.506 | 0.363 | 0.399 | 0.450 |
| VGAE | GB1 | 0.149 | -0.087 | 0.045 | 0.078 | -0.054 | 0.049 | 0.250 | -0.121 | 0.098 | 0.412 | -0.156 | 0.105 | 0.423 | -0.203 | 0.081 | 0.577 | -0.172 | 0.150 |
| | GB2 | 0.392 | -0.075 | **0.007** | 0.185 | -0.045 | **0.023** | 0.418 | -0.180 | **0.017** | 0.781 | -0.218 | **0.030** | 0.522 | -0.386 | **0.009** | 0.766 | -0.400 | **0.042** |
| | MI | 0.015 | 0.032 | 0.546 | 0.039 | 0.029 | 0.568 | 0.061 | 0.008 | 0.452 | 0.322 | 0.441 | 0.539 | -0.009 | -0.000 | 0.552 | 0.142 | 0.162 | 0.561 |
| DGI | GB1 | 0.218 | -0.118 | 0.060 | 0.105 | -0.084 | 0.082 | 0.023 | -0.055 | 0.254 | 0.515 | -0.196 | 0.326 | -0.009 | -0.012 | 0.511 | 0.119 | -0.400 | 0.277 |
| | GB2 | 0.283 | -0.161 | **0.053** | 0.149 | -0.122 | **0.056** | 0.029 | -0.043 | **0.182** | 0.399 | -0.299 | **0.288** | -0.087 | -0.373 | **0.491** | 0.216 | -0.449 | **0.273** |
| | MI | 0.076 | 0.007 | 0.536 | 0.102 | 0.010 | 0.475 | 0.222 | 0.096 | 0.513 | 0.254 | 0.132 | 0.535 | 0.016 | -0.185 | 0.511 | 0.112 | 0.020 | 0.594 |
| GRACE | GB1 | 0.142 | -0.057 | **0.016** | 0.113 | -0.030 | **0.062** | 0.182 | -0.016 | 0.158 | 0.338 | -0.149 | 0.022 | 0.124 | -0.262 | **0.155** | 0.253 | -0.276 | **0.046** |
| | GB2 | 0.155 | -0.071 | 0.017 | 0.140 | -0.028 | 0.063 | 0.235 | -0.041 | **0.052** | 0.382 | -0.154 | **0.055** | 0.012 | -0.443 | 0.217 | 0.133 | -0.382 | 0.151 |

et al., 2021). Actor is a graph where each node is an actor, and an edge indicates that two actors co-occur on a Wikipedia page (Tang et al., 2009). Furthermore, we also experiment with the DBpedia50k knowledge graph (Shi & Weninger, 2018), a subset of the DBpedia knowledge graph (Auer et al., 2007). The DBpedia50k graph does not contain node features; therefore, for this dataset we also train input node embeddings for the GNN. Statistics of all datasets is presented in Table 1.

## 5.4 Results

We present the results of the fidelity (Equation (7) and Equation (8)) and effect overlap (Equation (9)) metrics in Tables 2 for the homophilic and heterophilic graphs, and Table 3 for DBpedia50k. We denote GNNExplainer as MI, directly using the gradient as GB1, and Integrated Gradients as GB2.

**GB explanations are actionable.** The values of $\text{Fid}_a$ and $\text{Fid}_b$ for GB methods show that across all unsupervised learning methods and datasets, keeping edges above the explanation threshold always results in an increase of the similarity score, while keeping the edges below the threshold always results in a lower score. This means that GB explanations are **actionable**, as they allow interventions that result in a predictable effect on the similarity score. Relying on these explanations would allow to determine what edges contribute to increase (or decrease) in the score, and to interact with them by re-computing the similarity score with the knowledge provided by the explanation. This property is not observed with GNNExplainer, where the effect of keeping edges above the threshold is not clear, and certain patterns seem to depend on factors such as the model used to learn the embeddings, and the dataset. For example, for GAE and VGAE embeddings, keeping the edges above the threshold increases the similarity score more than keeping the edges below the threshold on Cora and Citeseer, but the opposite happens in the remaining datasets.

Table 3: Results of fidelity metrics ($\text{Fid}_a$ and $\text{Fid}_b$) and effect overlap (EO, lower is better) when applying different explanation methods to multiple unsupervised learning methods on the DBpedia50k knowledge graph.

| Method | | DBpedia50k | | |
|---|---|---|---|---|
| | | $\text{Fid}_a$ | $\text{Fid}_b$ | EO |
| GAE | MI | 0.057 | -0.073 | 0.564 |
| | GB1 | 0.148 | -0.190 | 0.050 |
| | GB2 | 0.149 | -0.213 | **0.028** |
| VGAE | MI | 0.059 | -0.054 | 0.614 |
| | GB1 | 0.149 | -0.185 | 0.059 |
| | GB2 | 0.182 | -0.187 | **0.037** |
| DGI | MI | -0.035 | -0.044 | 0.618 |
| | GB1 | 0.107 | -0.189 | 0.065 |
| | GB2 | 0.121 | -0.215 | **0.030** |
| GRACE | MI | -0.120 | -0.002 | 0.541 |
| | GB1 | 0.055 | -0.071 | **0.043** |
| | GB2 | 0.033 | -0.081 | 0.046 |

**GB explanations are consistent.** GB methods result in the lowest effect overlap across all learning methods and datasets. In the majority of the cases the overlap is around 0.1 or lower, indicating that the effect of keeping edges above the threshold is distinct from the effect of keeping the edges below the threshold, thus showing that GB explanations are **consistent**. Interestingly, this behavior is not as clear when using DGI embeddings on the heterophilic datasets (Chameleon, Actor, and Squirrel), where the overlap increases. This could be an effect of how the performance of DGI degrades in heterophilic graphs (Xiao et al., 2022), lowering the quality of its embeddings in graphs with these properties and thus becoming sensitive to the interventions required to compute the fidelity and effect overlap metrics. In the case of GNNExplainer, in the majority of the cases, the effect overlap is around 0.4 or even larger than 0.5, indicating that in almost half of the cases keeping the edges above the threshold increases the score, and in the other half the score decreases. We thus cannot rely on its explanations for a consistent effect on similarity scores.

We present further extended results in Appendix A, where we experiment with four additional GNN architectures. These results are consistent with our prior observations, while also showing that in some cases overlap can be low while actionability is limited, indicating the importance of assessing jointly our proposed metrics for explaining node similarities.

**Sparse GB explanations preserve effects.** We next evaluate whether explanations preserve their properties when made increasingly sparse. Since previous experiments showed that Integrated Gradients yields actionable and consistent explanations, we focus our study on this method.

To carry out this study, instead of taking all values of the explanation matrix above the threshold (as outlined in Eqs. 3 and 4), we drop a fraction $s$ of the smallest values in $\mathbf{M}_a$, and a fraction $s$ of the largest values in $\mathbf{M}_b$, where $s$ is the sparsity level taking values in the interval $[0, 1]$. When $s = 0$ all values in the explanation matrix are used, and we obtain the results previously described in Table 2. As $s$ increases, only the edges with the largest or the smallest values are kept in $\mathbf{M}_a$ and $\mathbf{M}_b$.

We compute the fidelity and effect overlap metrics for different values of sparsity from 0 up to 0.9 with increments of 0.1, when using GAE to learn embeddings. The results are shown in Fig. 2. We observe that the actionable and consistent properties of GB explanations remain almost constant across all datasets. This implies that when obtaining GB explanations, we can further reduce the set of edges in the explanation by up to 90%, and the different effects on the similarity scores will be preserved. This is beneficial for applications in which a more compact explanation is desired.

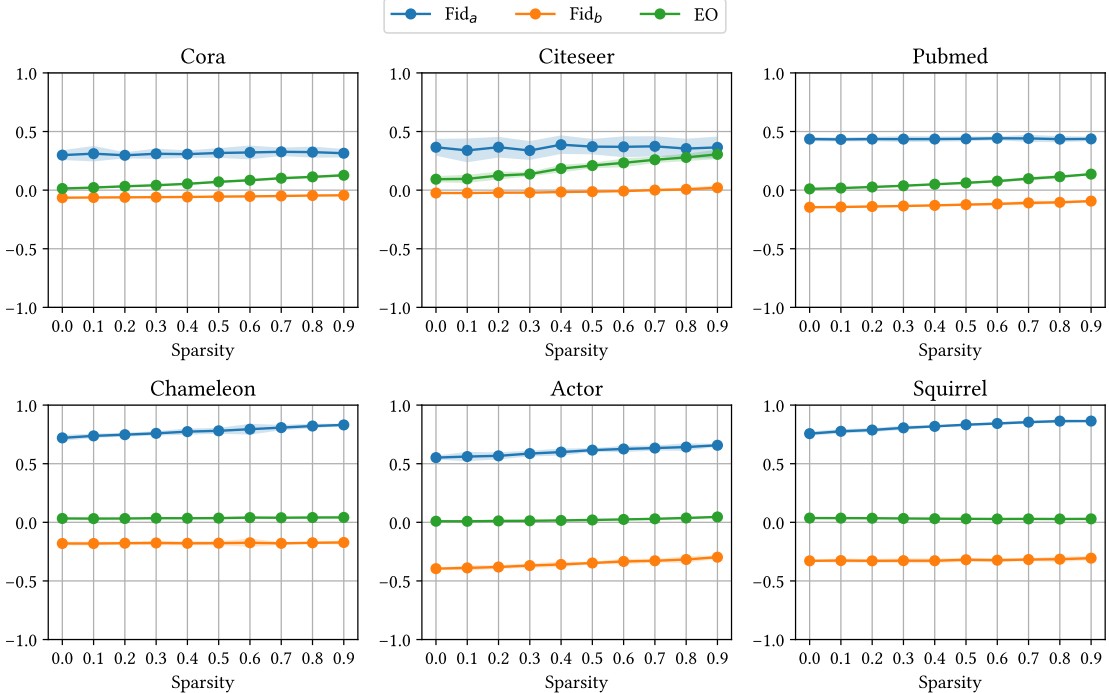

Figure 2: Influence of sparse explanations on fidelity metrics ($\text{Fid}_a$ and $\text{Fid}_b$) and effect overlap (EO), evaluated with GAE embeddings across different datasets. At zero sparsity, all edges above (or below) the explanation threshold are kept and used to compute the change in similarity scores $\text{Fid}_a$ (or $F_b$), as well as the effect overlap (EO). Larger values of sparsity indicate the fraction of edges discarded before computing the change in similarity scores. Confidence intervals are shown indicating two standard deviations over 10 runs.

**Examples.** We present concrete examples of the explanations obtained by GNNExplainer and Integrated Gradients in Fig. 3. For this case study, we train node embeddings using GAE on the DBpedia50k knowledge graph (Shi & Weninger, 2018). We then select the most relevant edges according to the explanation values assigned by each method. We consider two entities in the graph: *Lilium* and *Dendrobium*, which are two genera of flowering plants. Their similarity is reflected in a cosine similarity value of 0.705. We denote the effect attributed to each edge with colors, with blue indicating an increase in the similarity, red a decrease, and gray indicating little or no effect. When we obtain explanations with GNNExplainer, we observe that a few edges increase the similarity score, and none of them are in the 1-hop neighborhood of the entities, where their similarities are apparent. Both entities belong to the *Plant* kingdom and the *Flowering Plant* division. With gradient-based explanations, we observe that edges containing this information contribute to the increase of the similarity score, with the highest contributions (illustrated with the thickness of the edges) assigned to the relationships with *Plant* and *Flowering Plant*. Conversely, this analytical framework also enables the identification of specific informational elements that exert a negative influence on the calculated similarity score, thereby facilitating diagnostic assessment of dissimilarity factors. Overall, we note that gradient-based explanations are intuitive, by indicating both the magnitude and direction in which inputs affect similarity scores.

## 6 Conclusion

We have studied the problem of explaining node similarities computed by graph neural networks, a setting relevant in practice but largely overlooked by prior work on GNN explainability. Building on principles from explainable AI, we introduced criteria that explanations of similarity should satisfy, and we operationalized them through intervention-based metrics. We have demonstrated the utility of our framework by applying

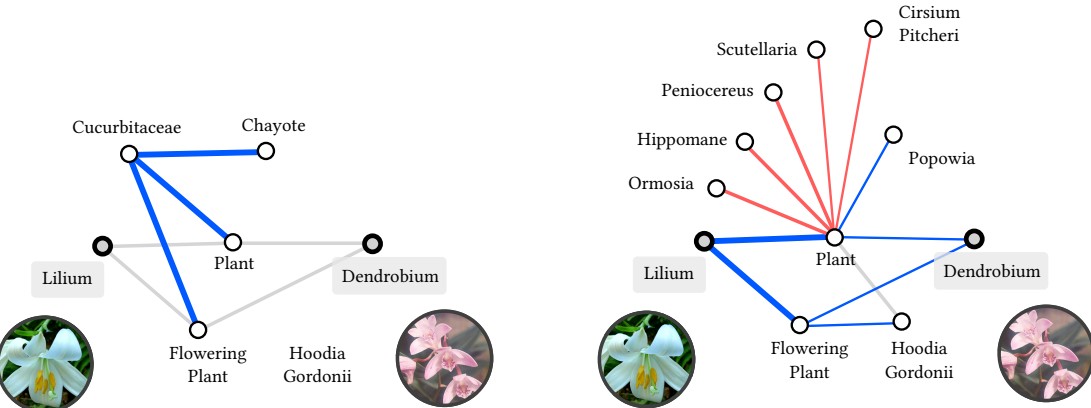

(a) GNNExplainer explanation.

(b) Gradient-based explanation.

Figure 3: Example of explanations provided by GNNExplainer (3a) and Integrated Gradients (3b) for the similarity computed between two entities in the DBpedia50k knowledge graph: *Lilium* and *Dendrobium*, two genera of flowering plants. Edge thickness indicates magnitude, while the color of the edges is associated with the score, i.e., blue edges reflect the score increase, red edges decrease, and gray edges have little effect.

it to representative mutual-information and gradient-based methods. Our results reveal that unlike prior results on supervised tasks like node classification (Ying et al., 2019; Luo et al., 2020; Yuan et al., 2023), gradient-based methods are more suitable in the setting of node similarity, by providing explanations with a predictable and consistent effect of increasing or decreasing similarity scores. Furthermore, we observe that the complexity of the explanations can be reduced while maintaining their desirable properties.

In this work, we focused on evaluating explanation methods based on how graph interventions derived from them affect similarity scores. An additional important aspect to be taken into account in the practical choice of an explanation method is its computational complexity. Gradient-based explanations require computing the gradient of a scalar similarity score with respect to the input graph, which amounts to a single backward pass through the GNN and scales linearly with the size of the graph up to constant factors. In contrast, mutual-information–based methods such as GNNExplainer rely on an iterative optimization procedure that repeatedly evaluates and differentiates the GNN over multiple steps, making them strictly more expensive in practice. While computational cost was not the main focus of our empirical study, it represents an additional criterion that may further favor gradient-based explanations in large-scale similarity search settings.

Beyond the specific methods examined here, our criteria and evaluation methodology offer a general foundation for evaluating explanations of similarity on graphs. They can inform the development of new methods for node similarity, and the design of methods that are explainable *a priori*. Moreover, based on our proposed methodology, further explainers (e.g., PGExplainer Luo et al. (2020), SubgraphX Yuan et al. (2021), PGM-Explainer Vu & Thai (2020)) could be analyzed, which is an interesting direction left for future research.

While we focused on cosine similarity, as it is the standard choice for fast dense similarity search over learned embeddings, our proposed intervention-based metrics are not specific to cosine similarity. They only require a similarity function defined over node embeddings, and can be applied to alternatives such as Euclidean distance without modification. The intervention logic, which measures how explanations predictably affect similarity under controlled graph perturbations, remains the same. We expect the qualitative behavior observed in our study to extend to other commonly used similarity measures without modification. We view these directions as promising avenues for future work.

## Acknowledgments

For this work, Michael Cochez is partially funded by the Elsevier Discovery Lab, partially funded by the Graph-Massivizer project, funded by the Horizon Europe programme of the European Union (grant 101093202), and supported by a gift from Accenture LLP. His work on this publication is in part based upon work from COST Action CA23147 GOBLIN - Global Network on Large-Scale, Cross-domain and Multilingual Open Knowledge Graphs, and COST Action CA24121 - Knowledge Graphs in the Era of Large Language Models (KGELL), both supported by COST (European Cooperation in Science and Technology, https://www.cost.eu). Daniel Daza performed a large portion of this work during an internship at the Bosch Center for Artificial Intelligence, Germany. Besides, he was funded by the Elsevier discovery lab and a gift from Accenture LLP.

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

Table 4: Results of fidelity metrics ($\text{Fid}_a$ and $\text{Fid}_b$) and effect overlap (EO, lower is better) when using an SGC (Wu et al., 2019) as the base GNN architecture.

| Method | | Cora | | | Citeseer | | | Pubmed | | | Chameleon | | | Actor | | | Squirrel | | |
| | | $\text{Fid}_a$ | $\text{Fid}_b$ | EO | $\text{Fid}_a$ | $\text{Fid}_b$ | EO | $\text{Fid}_a$ | $\text{Fid}_b$ | EO | $\text{Fid}_a$ | $\text{Fid}_b$ | EO | $\text{Fid}_a$ | $\text{Fid}_b$ | EO | $\text{Fid}_a$ | $\text{Fid}_b$ | EO |
|---|---|---|---|---|---|---|---|---|---|---|---|---|---|---|---|---|---|---|---|
| | MI | 0.020 | -0.005 | 0.916 | 0.185 | 0.006 | 0.832 | 0.090 | 0.231 | 0.978 | 0.231 | -0.069 | 0.668 | 0.161 | -0.053 | 0.676 | 0.042 | 0.250 | 0.892 |
| GAE | GB1 | 0.049 | -0.036 | **0.067** | 0.075 | -0.048 | 0.050 | 0.214 | -0.008 | 0.232 | 0.137 | -0.191 | **0.034** | 0.253 | -0.254 | **0.052** | 0.281 | -0.005 | 0.251 |
| | GB2 | 0.060 | -0.040 | 0.074 | 0.239 | -0.033 | **0.049** | 0.433 | -0.120 | **0.012** | 0.185 | -0.169 | 0.117 | 0.209 | -0.221 | 0.095 | 0.561 | -0.233 | **0.010** |
| | MI | 0.072 | 0.135 | 0.957 | 0.179 | 0.010 | 0.880 | 0.066 | 0.265 | 0.797 | 0.312 | -0.041 | 0.678 | 0.225 | -0.044 | 0.733 | -0.014 | 0.346 | 0.516 |
| VGAE | GB1 | 0.177 | -0.060 | 0.105 | 0.098 | -0.044 | 0.076 | 0.261 | -0.043 | 0.206 | 0.233 | -0.283 | **0.030** | 0.362 | -0.387 | **0.039** | 0.355 | 0.013 | 0.307 |
| | GB2 | 0.539 | -0.063 | **0.015** | 0.280 | -0.035 | **0.072** | 0.479 | -0.166 | **0.017** | 0.289 | -0.245 | 0.065 | 0.315 | -0.324 | 0.098 | 0.603 | -0.324 | **0.018** |
| | MI | 0.018 | 0.000 | 0.955 | -0.079 | 0.005 | **0.350** | -0.089 | -0.027 | **0.417** | -0.010 | 0.134 | 0.374 | 0.017 | 0.042 | 0.479 | 0.008 | -0.015 | 0.823 |
| DGI | GB1 | 0.017 | -0.010 | 0.015 | -0.013 | -0.104 | 0.353 | -0.112 | -0.096 | 0.914 | 0.209 | -0.213 | **0.110** | 0.088 | -0.053 | **0.055** | -0.208 | -0.287 | 0.682 |
| | GB2 | 0.018 | -0.011 | **0.006** | -0.085 | -0.338 | 0.527 | -0.110 | -0.218 | 0.553 | 0.310 | -0.261 | 0.125 | 0.079 | -0.059 | 0.068 | -0.265 | -0.403 | **0.650** |
| | MI | 0.007 | -0.005 | 0.837 | 0.010 | -0.002 | 0.908 | 0.118 | 0.106 | 0.729 | 0.140 | -0.017 | 0.813 | -0.066 | -0.064 | 0.953 | 0.075 | -0.008 | 0.608 |
| GRACE | GB1 | 0.047 | -0.040 | 0.074 | 0.030 | -0.021 | 0.095 | 0.115 | -0.045 | 0.162 | 0.098 | -0.107 | **0.033** | 0.095 | -0.244 | **0.027** | 0.070 | -0.064 | 0.262 |
| | GB2 | 0.057 | -0.047 | **0.050** | 0.041 | -0.024 | **0.085** | 0.221 | -0.060 | **0.045** | 0.124 | -0.104 | 0.047 | 0.015 | -0.321 | 0.200 | 0.080 | -0.189 | **0.185** |

# A    Additional results

Our main experimental results aim to investigate the effect of the metrics we proposed on different embedding learning algorithms and datasets when the architecture is designed from GCN layers (Kipf & Welling, 2017).

To determine whether our results generalize to other architectures, here we present extended results where we repeat the experiments using four additional architectures: SGC (Wu et al., 2019), a simplified variant of the GCN that reformulates message passing in multi-layer GNNs as a single matrix multiplication with powers of the adjacency matrix; GraphSAGE (Hamilton et al., 2017b), which for a given node, concatenates embeddings of the node from previous layers during the neighborhood aggregation process (in contrast to the GCN which mixes only embeddings from the same layer); MixHop (Abu-El-Haija et al., 2019), which is designed to capture higher-order patterns by computing at each layer representations from powers of the adjacency matrix; and R-GCN (Schlichtkrull et al., 2018), a variant of the GCN that learns different weight matrices for heterogeneous relation types that arise in knowledge graphs.

We present results with SGC in Table 4, GraphSAGE in Table 5, and MixHop in Table 6. On the DBPedia50k knowledge graph, we present additional results using these three architectures plus the R-GCN in Table 7. The extended results indicate that the behaviour observed with our proposed metrics remains under changes in the architecture in the majority of the cases: gradient-based explanations are actionable, as interventions based on them produce predictable effects on the score (either increasing it or decreasing it); and they are consistent, as these effects are distinct, which is demonstrated by the low effect overlap.

A special case is found in the combination of DGI as a learning algorithm, and GraphSAGE as the base GNN (shown in Table 5). Here we note that in virtually all cases, any intervention results in *decreasing* the similarity between two nodes (denoted by the negative values of $\text{Fid}_a$ and $\text{Fid}_b$) but the magnitude of changes is larger when using gradient-based explanations, resulting in higher actionability. In contrast, the mutual information (MI) method results in smaller changes, thereby indicating low actionability. In spite of this, the MI method results in lower overlap in several cases, indicating higher consistency than gradient-based methods.

We conclude that EO must be interpreted jointly with fidelity: the lower EO of MI here follows from near-zero effects, whereas gradient-based explanations result in substantially stronger interventions (higher actionability) even though their effects are not sign-separated, which raises EO. Despite occasional cases where MI attains lower EO, gradient-based explanations are preferable for similarity explanation because they provide stronger, controllable interventions; MI's improved overlap in this regime is largely explained by its limited impact on the similarity score. This suggests that on DGI+GraphSAGE, the similarity function is globally fragile (interventions mostly reduce similarity), making sign-based separation difficult for any explainer despite differences in actionability.

Table 5: Results of fidelity metrics ($\text{Fid}_a$ and $\text{Fid}_b$) and effect overlap (EO, lower is better) when using GraphSAGE (Hamilton et al., 2017b) as the base GNN architecture.

| Method | | Cora $\text{Fid}_a$ | $\text{Fid}_b$ | EO | Citeseer $\text{Fid}_a$ | $\text{Fid}_b$ | EO | Pubmed $\text{Fid}_a$ | $\text{Fid}_b$ | EO | Chameleon $\text{Fid}_a$ | $\text{Fid}_b$ | EO | Actor $\text{Fid}_a$ | $\text{Fid}_b$ | EO | Squirrel $\text{Fid}_a$ | $\text{Fid}_b$ | EO |
|---|---|---|---|---|---|---|---|---|---|---|---|---|---|---|---|---|---|---|---|
| GAE | MI | 0.000 | 0.002 | 0.496 | 0.001 | 0.000 | 0.724 | 0.081 | 0.074 | 0.908 | 0.036 | 0.029 | 0.720 | 0.060 | 0.031 | 0.810 | 0.037 | 0.037 | 0.969 |
| | GB1 | 0.012 | -0.008 | 0.159 | 0.024 | -0.022 | 0.149 | 0.171 | -0.041 | 0.234 | 0.082 | -0.038 | 0.103 | 0.094 | -0.035 | 0.235 | 0.144 | -0.083 | 0.113 |
| | GB2 | 0.019 | -0.014 | **0.080** | 0.030 | -0.026 | **0.083** | 0.264 | -0.130 | **0.046** | 0.123 | -0.050 | **0.070** | 0.194 | -0.075 | **0.105** | 0.188 | -0.121 | **0.028** |
| VGAE | MI | 0.057 | 0.061 | 0.892 | 0.056 | 0.058 | 0.832 | 0.079 | 0.121 | 0.944 | 0.041 | 0.035 | 0.898 | 0.097 | 0.085 | 0.967 | 0.035 | 0.043 | 0.919 |
| | GB1 | 0.116 | -0.037 | 0.103 | 0.091 | -0.021 | 0.137 | 0.202 | -0.020 | 0.269 | 0.151 | -0.082 | 0.106 | 0.232 | -0.077 | 0.177 | 0.158 | -0.106 | 0.116 |
| | GB2 | 0.183 | -0.058 | **0.026** | 0.168 | -0.042 | 0.082 | 0.343 | -0.142 | 0.053 | 0.257 | -0.126 | **0.022** | 0.338 | -0.157 | **0.068** | 0.208 | -0.154 | **0.030** |
| DGI | MI | -0.105 | -0.236 | **0.720** | -0.038 | -0.365 | **0.626** | 0.005 | -0.000 | 0.969 | 0.000 | -0.027 | **0.708** | 0.013 | -0.009 | **0.373** | -0.038 | -0.017 | **0.242** |
| | GB1 | -0.303 | -0.410 | 0.946 | -0.464 | -0.423 | 0.878 | -0.651 | -0.604 | 0.970 | -0.276 | -0.433 | 0.869 | -0.133 | -0.396 | 0.634 | -0.148 | -0.218 | 0.845 |
| | GB2 | -0.475 | -0.660 | 0.959 | -0.420 | -0.783 | 0.672 | -0.170 | -0.213 | **0.953** | -0.385 | -0.627 | 0.901 | -0.317 | -0.556 | 0.807 | -0.251 | -0.365 | 0.862 |
| GRACE | MI | 0.001 | -0.002 | 0.754 | -0.004 | -0.002 | 0.676 | -0.013 | -0.004 | 0.459 | 0.027 | -0.002 | 0.576 | 0.016 | -0.005 | 0.556 | -0.001 | -0.000 | 0.645 |
| | GB1 | 0.023 | -0.024 | 0.079 | 0.010 | -0.016 | 0.075 | 0.036 | -0.056 | 0.087 | 0.050 | -0.043 | 0.057 | 0.054 | -0.064 | 0.045 | 0.080 | -0.108 | 0.085 |
| | GB2 | 0.029 | -0.028 | **0.046** | 0.011 | -0.017 | **0.066** | 0.048 | -0.061 | **0.026** | 0.071 | -0.050 | **0.035** | 0.066 | -0.077 | **0.026** | 0.107 | -0.146 | **0.026** |

Table 6: Results of fidelity metrics ($\text{Fid}_a$ and $\text{Fid}_b$) and effect overlap (EO, lower is better) when using MixHop (Abu-El-Haija et al., 2019) as the base GNN architecture.

| Method | | Cora $\text{Fid}_a$ | $\text{Fid}_b$ | EO | Citeseer $\text{Fid}_a$ | $\text{Fid}_b$ | EO | Pubmed $\text{Fid}_a$ | $\text{Fid}_b$ | EO | Chameleon $\text{Fid}_a$ | $\text{Fid}_b$ | EO | Actor $\text{Fid}_a$ | $\text{Fid}_b$ | EO | Squirrel $\text{Fid}_a$ | $\text{Fid}_b$ | EO |
|---|---|---|---|---|---|---|---|---|---|---|---|---|---|---|---|---|---|---|---|
| GAE | MI | 0.002 | -0.002 | 0.558 | 0.013 | 0.003 | 0.604 | 0.080 | 0.165 | 0.912 | 0.069 | -0.098 | 0.860 | 0.029 | -0.038 | 0.932 | 0.057 | 0.048 | 0.878 |
| | GB1 | 0.047 | -0.038 | 0.065 | 0.027 | -0.016 | **0.098** | 0.198 | 0.047 | 0.428 | 0.113 | -0.202 | **0.043** | 0.170 | -0.198 | **0.052** | 0.239 | -0.023 | 0.294 |
| | GB2 | 0.055 | -0.042 | **0.059** | 0.032 | -0.018 | 0.117 | 0.363 | -0.144 | **0.050** | 0.114 | -0.226 | 0.071 | 0.152 | -0.222 | 0.060 | 0.272 | -0.230 | **0.033** |
| VGAE | MI | 0.011 | -0.011 | 0.990 | 0.006 | -0.004 | 0.908 | 0.088 | 0.279 | 0.942 | 0.112 | 0.086 | 0.921 | 0.171 | 0.130 | 0.800 | 0.101 | 0.111 | 0.784 |
| | GB1 | 0.079 | -0.070 | 0.072 | 0.033 | -0.029 | **0.148** | 0.248 | 0.026 | 0.355 | 0.268 | -0.130 | 0.062 | 0.380 | -0.119 | 0.119 | 0.272 | 0.035 | 0.420 |
| | GB2 | 0.083 | -0.075 | **0.064** | 0.038 | -0.027 | 0.167 | 0.425 | -0.180 | **0.057** | 0.361 | -0.170 | **0.021** | 0.499 | -0.247 | **0.024** | 0.376 | -0.291 | **0.031** |
| DGI | MI | -0.023 | 0.002 | **0.342** | -0.031 | 0.011 | **0.386** | 0.065 | 0.023 | 0.757 | 0.004 | 0.006 | 0.596 | -0.079 | -0.010 | **0.370** | 0.000 | -0.030 | **0.656** |
| | GB1 | -0.142 | -0.245 | 0.817 | -0.211 | -0.349 | 0.696 | -0.037 | -0.129 | 0.652 | -0.003 | -0.214 | **0.360** | -0.227 | -0.524 | 0.762 | -0.303 | -0.417 | 0.898 |
| | GB2 | -0.070 | -0.205 | 0.711 | -0.169 | -0.351 | 0.601 | -0.019 | -0.097 | **0.541** | -0.009 | -0.334 | 0.407 | -0.032 | -0.138 | 0.577 | -0.212 | -0.376 | 0.773 |
| GRACE | MI | -0.016 | -0.024 | 0.590 | -0.014 | -0.010 | 0.584 | -0.021 | -0.026 | 0.567 | -0.021 | -0.033 | 0.608 | -0.114 | -0.091 | 0.587 | -0.034 | -0.069 | 0.742 |
| | GB1 | 0.032 | -0.072 | **0.083** | 0.030 | -0.051 | 0.153 | 0.006 | -0.091 | 0.324 | 0.061 | -0.149 | **0.077** | 0.027 | -0.257 | **0.164** | 0.017 | -0.120 | **0.211** |
| | GB2 | 0.031 | -0.081 | 0.100 | 0.028 | -0.058 | **0.142** | 0.012 | -0.094 | **0.304** | 0.062 | -0.156 | 0.086 | -0.047 | -0.308 | 0.382 | 0.008 | -0.133 | 0.296 |

Table 7: Results of fidelity metrics ($\text{Fid}_a$ and $\text{Fid}_b$) and effect overlap (EO, lower is better) on the DBPedia50k dataset, when using SGC (Wu et al., 2019), GraphSAGE (Hamilton et al., 2017b), MixHop (Abu-El-Haija et al., 2019), and R-GCN (Schlichtkrull et al., 2018) as the base GNN architecture.

| Method | | SGC $\text{Fid}_a$ | $\text{Fid}_b$ | EO | GraphSAGE $\text{Fid}_a$ | $\text{Fid}_b$ | EO | MixHop $\text{Fid}_a$ | $\text{Fid}_b$ | EO | R-GCN $\text{Fid}_a$ | $\text{Fid}_b$ | EO |
|---|---|---|---|---|---|---|---|---|---|---|---|---|---|
| GAE | MI | 0.006 | -0.112 | 0.914 | 0.013 | -0.037 | 0.874 | -0.029 | -0.079 | 1.000 | -0.014 | -0.019 | 0.970 |
| | GB1 | 0.124 | -0.172 | **0.089** | 0.051 | -0.084 | 0.241 | 0.096 | -0.166 | 0.127 | 0.095 | -0.114 | 0.198 |
| | GB2 | 0.155 | -0.184 | 0.129 | 0.094 | -0.092 | **0.159** | 0.110 | -0.211 | **0.087** | 0.116 | -0.132 | **0.137** |
| VGAE | MI | -0.009 | 0.071 | 0.711 | 0.012 | -0.052 | 0.833 | 0.005 | -0.055 | 0.976 | -0.014 | -0.025 | 0.929 |
| | GB1 | 0.166 | -0.128 | 0.153 | 0.067 | -0.081 | 0.242 | 0.137 | -0.172 | 0.083 | 0.071 | -0.114 | 0.186 |
| | GB2 | 0.375 | -0.160 | **0.114** | 0.081 | -0.114 | **0.135** | 0.156 | -0.206 | **0.041** | 0.089 | -0.136 | **0.091** |
| DGI | MI | -0.037 | 0.273 | 0.252 | -0.087 | -0.017 | 0.753 | -0.082 | -0.020 | **0.239** | -0.041 | -0.054 | 0.850 |
| | GB1 | 0.264 | -0.080 | 0.140 | 0.123 | -0.324 | 0.357 | -0.146 | -0.213 | 0.843 | 0.057 | -0.171 | 0.331 |
| | GB2 | 0.345 | -0.113 | **0.095** | 0.093 | -0.508 | **0.234** | -0.095 | -0.211 | 0.612 | 0.064 | -0.215 | **0.230** |
| GRACE | MI | 0.010 | -0.022 | 0.813 | -0.001 | -0.007 | 0.546 | -0.015 | -0.010 | 0.373 | -0.022 | -0.024 | 0.813 |
| | GB1 | 0.108 | -0.083 | 0.058 | 0.017 | -0.024 | 0.074 | 0.021 | -0.043 | **0.085** | 0.077 | -0.072 | 0.203 |
| | GB2 | 0.107 | -0.089 | **0.048** | 0.019 | -0.030 | **0.046** | 0.017 | -0.045 | 0.099 | 0.057 | -0.072 | **0.201** |

