# OpenReview forum: "Explaining Graph Neural Networks for Node Similarity on Graphs"
_TMLR — Accepted by TMLR_

### Review · Reviewer_mZ83 · 2025-12-14

**Summary Of Contributions:**

This paper explores how to interpret the node similarity scores computed by GNNs. It proposes three core criteria for evaluating similarity explanations, including actionability, consistency, and sparsity. The paper operationalizes them via intervention-based metrics: fidelity, effect overlap, and sparsity preservation. The authors compared two explanatory methods (i.e., MI based methods and GB based methods) on six datasets (homogeneous graphs, heterogeneous graphs, and knowledge graphs) and four unsupervised node embedding methods (GAE, VGAE, DGI, and GRACE). The key finding is that gradient-based methods outperform MI-based methods in satsfying the three criteria.

**Additional Comments:**

NA

**Audience:**

Yes

**Audience Explanation:**

Yes, some individuals in TMLR's audience would be interested in the findings of this paper. However, the scope of interest may be limited to researchers who are specifically focused on interpretability methods for graph neural networks and unsupervised node embeddings. Its appeal may not extend to those working on more general graph learning or supervised based graph learning tasks.

**Claims And Evidence:**

No

**Claims Explanation:**

1. This paper relies on only GCN as the backbone for node embedding. However, this message passing architecture can not be effectively applied to heterogeneous graphs and knowledge graphs. The bad interpretation performance of MI may be due to the embedding learning process itself rather than the interpretability method. Therefore, the experimental results on heterogeneous graphs and knowledge graphs are not convincing.
2. Another thing is that different message passing methods create node embeddings with different properties. For example, GCN tends to create smooth node embeddings, while the Transformer based GNN (e.g., graph transformer [`1`]) and the GNNs motivated by Weisfeiler and Leman test (e.g., k-GNN [`2`] and MPSN [`3`] ) capture longer-range dependencies. The conclusions that GB performs better than MI drawn from GCN may not generalize to other message passing architectures.

---
**Reference**

[1] Rampášek, L., Galkin, M., Dwivedi, V. P., Luu, A. T., Wolf, G., & Beaini, D. (2022). Recipe for a general, powerful, scalable graph transformer. Advances in Neural Information Processing Systems, 35, 14501-14515.

[2] Morris, C., Ritzert, M., Fey, M., Hamilton, W. L., Lenssen, J. E., Rattan, G., & Grohe, M. (2019, July). Weisfeiler and leman go neural: Higher-order graph neural networks. In Proceedings of the AAAI conference on artificial intelligence (Vol. 33, No. 01, pp. 4602-4609).

[3] Bodnar, C., Frasca, F., Wang, Y., Otter, N., Montufar, G. F., Lio, P., & Bronstein, M. (2021, July). Weisfeiler and lehman go topological: Message passing simplicial networks. In International conference on machine learning (pp. 1026-1037). PMLR.

**Requested Changes:**

1. Please consider my concern raised above.
2. No explicit justification for threshold choices (t=0.5 for MI, t=0 for GB methods).
3. Why edge-only explanations are prioritized over node features? I think both edge and node feature explanations are not human-understandable.
4. While Section 2 categorizes prior work, it fails to clearly link gaps in existing literature to the specific contributions of this paper.

---

> ### Author Response · Authors · 2026-01-06
>
> We sincerely thank the reviewer for the careful reading of the manuscript and for raising these thoughtful and constructive concerns. We have incorporated the feedback in the latest version of our paper (changes are shown in green in the PDF). Below we address each point in detail.
>
> **Backbone architecture and heterogeneous / knowledge graphs:** We appreciate the reviewer’s concern regarding the reliance on GCNs, particularly in heterogeneous and knowledge graph settings. This point motivated an extension of our experimental study. In the revised version, we include four additional GNN architectures: SGC, GraphSAGE, MixHop, and R-GCN. Notably, R-GCN explicitly models typed relations and is designed for heterogeneous and knowledge graphs. The extended results (Appendix A, discussed in Section 5.4) show that the qualitative trends observed in the main paper persist across architectures: gradient-based explanations remain more actionable and consistent than mutual-information-based ones.
>
> **Generality beyond GCN-style message passing:** We fully agree that different message-passing paradigms induce different embedding properties. To address this, we include MixHop, which captures higher-order dependencies through powers of the adjacency matrix, going beyond standard one-hop smoothing. While we do not include graph transformers or higher-order GNNs (e.g., k-GNNs or MPSNs) due to computational constraints, we emphasize that our contribution lies in proposing architecture-agnostic evaluation criteria. The proposed metrics make no assumptions about the underlying message-passing scheme and can be directly applied to explanations produced by transformer-based or higher-order GNNs.
>
> **Threshold choices:** For MI-based methods, explanation values are interpreted as Bernoulli probabilities over edge selection, making 0.5 the natural decision boundary. For gradient-based methods, non-zero gradients indicate positive or negative influence on the similarity score, motivating a threshold of zero to distinguish increasing versus decreasing effects. We have included this explanation in sections 5.1.1 and 5.1.2.
>
> **Edge vs. node feature explanations:** We agree that both edge- and feature-level explanations can be challenging for human interpretation, which motivates measuring their quality as we aim in our work. We do not intend to prioritize edges over node features: both MI-based and gradient-based methods support feature attributions, and our metrics apply equally in that setting, as they depend only on changes in the similarity score under intervention. We focus on edges for concreteness and consistency across datasets, but the framework naturally extends to node features without modification.
>
> **Positioning with respect to prior work:** We have revised the related work section to more clearly articulate the gap addressed by our paper. Existing explainability methods primarily focus on supervised predictions, while data valuation methods analyze effects on parameters or classification performance. In contrast, our work targets the explanation of individual, continuous similarity scores in an unsupervised, label-free setting. This is an explanatory target that so far has not been directly addressed by prior approaches and motivates the need for our proposed new criteria and evaluation methods.

---

> > ### Comment · Reviewer_mZ83 · 2026-01-13
> > **Thanks for the response, the motivation and impact of this paper still remain unclear**
> >
> > Thanks for the response. I'm satisfied with the other clarifications, but I still have concerns about the motivation for this work.
> >
> > As I noted in my initial review, I'm concerned that **the audience for this work would be very narrow**. The paper analyzes graph XAI methods in a new setting, but it doesn't convincingly establish that this setting has clear practical needs or sufficient research value.
> >
> > My main question is: **why do we actually need explainability for unsupervised node similarity?** For supervised tasks like node classification or link prediction, the reasons for explainability are clear, for example, you want to help users trust/understand decisions. But for unsupervised similarity search, I don't think these motivations can apply in the same way.
> >
> > Unsupervised node representation learning aims to learn a good embedding space that can be used for downstream tasks (such as classification, clustering, and retrieval). Similarity is merely a byproduct of embedding space optimization.
> > From this perspective, explaining "why these two nodes are similar" is actually explaining the geometry of the embedding space, not a specific prediction task. This is a different kind of problem from explaining a classifier's prediction that "this node belongs to category A".
> >
> > However, if you need explainability, it's usually at the downstream level, not at the embedding level. So I'm left wondering: who is the intended user of these explanations? What decisions would they make differently after seeing them?

---

> > > ### Author Response · Authors · 2026-01-14
> > >
> > > We thank the reviewer for continuing to engage with the motivation of this work. We emphasize that the problem of explaining node similarity studied in this paper is driven by concrete and practically relevant applications. One representative example is the analysis of large graphs of companies, encoding relationships such as suppliers, competitors, and other business relations. In these settings, users are often interested in retrieving companies similar to a given one, for instance to identify potential suppliers with desired characteristics. However, such graphs are typically large and complex, and as a result, users often lack a priori knowledge of which criteria meaningfully define similarity. In this context, leveraging GNNs to identify similar entities, and, crucially, to explain why they are considered similar, is both relevant and impactful.
> > >
> > > Similar needs arise in a variety of other applications, including product recommendation in recommender systems, scientific discovery over biomedical graphs, graph-based recommendation, and exploratory data analysis. In these settings, similarity is not merely an auxiliary notion but a fundamental primitive that directly supports downstream procedures such as retrieval, clustering, and recommendation.
> > > Moreover, simply presenting similar nodes to users is insufficient. Instead, explanations are needed to answer concrete and actionable questions such as:
> > > - Why are these two nodes considered similar?
> > > - Which parts of the graph contribute to (or detract from) their similarity?
> > > - How could the graph be modified to increase or decrease this similarity?
> > >
> > > Our experiments demonstrate that the proposed metrics can meaningfully evaluate how effectively different explanation methods address these questions, by explicitly intervening on the graph and observing the resulting changes in similarity scores.
> > >
> > > Importantly, this motivation is not unique to our work. Recent survey papers, which we cite, explicitly identify explainability for unsupervised graph representation learning as an open and important research direction. For example, Liu et al. (TKDE 2022) highlight that many applications of unsupervised graph learning are risk-sensitive (e.g., fraud detection), making explainable representations practically important, while noting that most existing approaches remain black-box. Similarly, Xie et al. (TPAMI 2022) observe that the majority of existing explanation methods focus on supervised tasks and call for further work on explaining GNNs trained with unsupervised objectives. Our work directly addresses this gap by formalizing the problem of explaining node similarity in unsupervised GNNs and by proposing task-specific criteria and evaluation metrics. We have added a remark in the related work section to more explicitly reflect this motivation.
> > > We acknowledge that this topic may most strongly be of interest to researchers working on explainability, unsupervised graph learning, and similarity-based applications. Nevertheless, we believe that formalizing this problem and providing principled evaluation tools for this underexplored setting constitutes a meaningful contribution to the broader explainable AI and graph learning communities.
> > >
> > > **References**
> > >
> > > Liu, Y., et al. *Graph Self-supervised Learning: A Survey.* IEEE Transactions on Knowledge and Data Engineering, 2022.
> > >
> > > Xie, Y., et al. *Self-supervised Learning of Graph Neural Networks: A Unified Review.* IEEE Transactions on Pattern Analysis and Machine Intelligence, 2022.

---

### Review · Reviewer_y8Xx · 2025-12-14

**Summary Of Contributions:**

This manuscript explores an underexplored issue in the field of GNN interpretability: how to explain similarity between nodes. To this end, the authors propose an evaluation framework with three key criteria. Comparative experiments on multiple datasets between mutual-information-based and gradient-based methods yield certain insights.

**Strengths**
1) Explaining node similarity appears to be the first work of its kind, advancing the field.
2) The authors employ a wide range of datasets, including both homophilic and heterophilic graphs.
3) The authors highlight specific advantages of gradient-based methods over mutual-information-based methods, providing heuristic guidance for future research.
4) The proposed evaluation metrics are reasonable, particularly the consistency criterion, which effectively distinguishes between positive and negative contributions of explanations.

**Weaknesses**
1) In page 3, the authors claim that their proposed metrics are model-agnostic, but in the experiments, only GCN is used as the encoder. Could the authors employ more GNNs (e.g., SGC, APPNP, GraphSAGE, etc.) to verify whether explanations for node similarity are consistent or follow similar trends across models?
2) In the experiments, the authors only consider GNNExplainer as a representative mutual-information-based method. It is recommended that the authors include more methods (e.g., PGExplainer, SubgraphX, PGM-Explainer) to better support their arguments.
3) The authors show experimentally that gradient-based methods generally provide better explanations, but an analysis of computational overhead is lacking.
4) Interpretability is closely related to data attribution or data valuation. It is suggested that the authors discuss in the related work the connections between interpretability and relevant studies [1-5].

In summary, i think this is a very interesting paper. The authors have identified a meaningful and valuable research problem, but the experimental study lacks breadth. If the above issues can be adequately addressed, I am inclined to accept this manuscript.

[1] GraphSVX: Shapley Value Explanations for Graph Neural Networks. ECML, 2021.
[2] Characterizing the Influence of Graph Elements. ICLR, 2023.
[3] RGE: ARepulsive Graph Rectification for Node Classification via Influence. ICML, 2023.
[4] Precedence-Constrained Winter Value for Effective Graph Data Valuation. ICLR, 2025.
[5] Exact Computation of Any-Order Shapley Interactions for Graph Neural Networks. ICLR, 2025.

**Audience:**

Yes

**Audience Explanation:**

This manuscript explores an underexplored issue in the field of GNN interpretability. I believe this will advance interpretability in areas such as recommendation systems, community detection, epidemic analysis, and complex networks.

**Claims And Evidence:**

Yes

**Claims Explanation:**

The authors propose three critical metrics and conduct relatively comprehensive experiments on multiple datasets.

**Requested Changes:**

Please refer  to **Weaknesses**.

---

> ### Author Response · Authors · 2026-01-06
>
> We thank the reviewer for the constructive feedback and the positive assessment of the paper’s contributions. We have incorporated it in the latest version of our paper (shown in green in the PDF). Below we address each concern in detail.
>
> **Additional GNN architectures:** We agree that demonstrating model-agnosticity beyond GCNs is important. In response, we have extended our experimental evaluation to four additional GNN architectures: SGC, GraphSAGE, MixHop, and R-GCN. The extended results, reported in Appendix A and discussed in Section 5.4, show that the qualitative trends identified in the main paper remain the same across architectures: gradient-based explanations remain more actionable and consistent than mutual-information-based ones. Importantly, these results also reveal regimes in which effect overlap alone can be misleading (e.g., low overlap but limited actionability), demonstrating the need to jointly assess the proposed metrics.
>
> **Scope of explanation methods:** Our primary goal is to introduce and validate evaluation criteria for explanations of node similarity, rather than to exhaustively benchmark all existing explainers. We demonstrate the utility of our metrics by applying them to representative methods from two fundamentally different explanation paradigms (mutual information and gradient-based) across a large experimental space spanning multiple datasets, learning objectives, and GNN architectures. Given the already substantial computational cost of this evaluation, we view extending the analysis to additional explainers (e.g., PGExplainer, SubgraphX, PGM-Explainer) as an important direction for future work, enabled by the framework introduced here.
>
> **Computational overhead:** We agree that computational complexity is an important practical consideration. We have therefore added a discussion in the conclusion clarifying that gradient-based explanations require a single backward pass to compute the gradient of a scalar similarity score with respect to the input graph, scaling linearly with graph size up to constant factors. In contrast, mutual information methods such as GNNExplainer rely on iterative optimization involving repeated forward and backward passes, making them strictly more expensive in practice. While computational cost was not the focus of our empirical study, it constitutes an additional criterion that may become relevant in general in large-scale similarity search settings.
>
> **Relation to data valuation:** We have expanded the related work to explicitly discuss the relationship between explainability and data valuation, citing recent works on graph influence and valuation. We clarify that data valuation methods typically study the effect of graph components on model parameters or supervised performance (e.g., classification accuracy), whereas our work focuses on explaining individual similarity computations in an unsupervised, label-free setting. As such, our objective is complementary and addresses a distinct explanatory target that existing data valuation approaches are not designed to capture.

---

> > ### Comment · Reviewer_y8Xx · 2026-01-25
> >
> > Thank you for the efforts of the authors, as most of my concerns have been adequately addressed.

---

### Review · Reviewer_5HUx · 2025-12-26

**Summary Of Contributions:**

This paper studies the problem of explaining node similarity scores produced by Graph Neural Networks trained in an unsupervised manner. While most prior work on explainable GNNs focuses on supervised tasks such as node classification, this submission correctly identifies that similarity search constitutes a different setting, where predictions are continuous and sensitive to small changes in graph structure. The authors analyze two prominent families of explanation methods, namely mutual information–based approaches and gradient-based approaches, and assess their suitability for explaining cosine similarity between node embeddings. They introduce three desiderata tailored to similarity explanations—actionability, consistency, and sparsity—and propose quantitative metrics to evaluate them through graph interventions. Through extensive experiments across multiple datasets and unsupervised learning methods, the paper shows that gradient-based explanations satisfy these desiderata more reliably than mutual information–based methods, which often yield ambiguous or inconsistent effects. A key strength of the paper is the clarity with which it motivates why relevance-based explanations are insufficient for similarity tasks, as well as the breadth of the experimental evaluation. A limitation is that the analysis is restricted to cosine similarity and edge-level explanations, which narrows the generality of the conclusions.

**Audience:**

Yes

**Audience Explanation:**

The findings of this paper are likely to be of interest to a significant portion of the TMLR audience, particularly researchers working on graph representation learning, unsupervised learning, and explainable machine learning. Similarity search on graphs is a foundational task with applications in recommendation systems, information retrieval, and knowledge graphs, yet explainability in this setting has received limited attention. By highlighting the mismatch between common explanation paradigms and similarity-based objectives, the paper provides insights that are broadly relevant beyond the specific methods evaluated. The work also offers practical guidance on which explanation techniques are more appropriate for similarity tasks, which is valuable for both researchers and practitioners designing explainable graph-based systems.

**Broader Impact Concerns:**

The paper does not raise immediate ethical concerns.

**Claims And Evidence:**

Yes

**Claims Explanation:**

The claims made in the paper are supported by clear and systematic empirical evidence. The authors operationalize their proposed desiderata using well-defined metrics based on controlled interventions on the graph and consistently apply these metrics across multiple datasets, graph structures, and unsupervised learning methods. The experimental results strongly support the central claim that gradient-based explanations are actionable and consistent, while mutual information–based explanations are not well suited to similarity tasks. The inclusion of both homophilic and heterophilic graphs, as well as a knowledge graph without node features, further strengthens the evidence by demonstrating robustness across different graph regimes. Qualitative examples complement the quantitative results and illustrate how gradient-based explanations align with intuitive semantic interpretations. While the evaluation focuses on a specific similarity function, within this scope the evidence is convincing and clearly presented.

**Requested Changes:**

While the paper is generally strong, it would benefit from a more explicit discussion of how the conclusions depend on the choice of cosine similarity. In particular, it would be valuable to clarify whether the proposed desiderata and the empirical findings are expected to hold for alternative similarity or distance measures, such as dot product similarity or Euclidean distance. Including such a discussion, or a limited comparison, would help assess the generality of the conclusions and strengthen the overall contribution, although this change is not strictly critical for acceptance.

---

> ### Author Response · Authors · 2026-01-06
>
> We thank the reviewer for the thoughtful comments and suggestions, and the positive assessment of the paper’s contributions.
>
> **On the similarity function:** In this work, we focus on cosine similarity because it is a common method for fast dense similarity search over learned embeddings, and it is widely used in graph representation learning, information retrieval, and knowledge graph applications [1,2]. Focusing on this established function also allows us to run the large scale experiments we consider, which comprise several datasets, embedding algorithms, and GNN architectures.
>
> Importantly, our proposed criteria and metrics for explainability are not tied to cosine similarity. They only require a similarity function defined over node embeddings, and can be applied to alternatives such as Euclidean distance without modification. The intervention logic, which measures how explanations predictably affect similarity under controlled graph perturbations, remains the same. We have now included a discussion of this important point in the Conclusion (changes are shown in green in the PDF).
>
> **References**
>
> [1] L. Shimomura et al. (2021), “A survey on graph-based methods for similarity searches in metric spaces”
>
> [2] Johnson et al. (2017). “Billion-scale similarity search with GPUs”.

---

### Decision · Action_Editor_ZRvo · 2026-02-11

**Recommendation:** Accept as is

**Additional Comments:**

This submission was reviewed by three expert reviewers. One reviewer argued that the claims made in the submission were not supported by sufficiently convincing evidence since GCN was the only backbone used to produce node representations. The authors conducted additional experiments during the revision, where they evaluated four additional architectures. The same trends were observed, and the reviewer's concern was adequately addressed. The reviewers also requested some additional changes, including: (i) a discussion of whether the empirical findings are likely to hold for other similarity or distance measures, (ii) a justification of the chosen values for the hyperparameter $t$, (iii) improvements to the related work section, (iv) the inclusion of more mutual-information-based methods in the experimental evaluation, and (v) a discussion of the computational complexity of the different types of explanation methods. Most of these concerns were addressed by the authors. After the revision, one reviewer recommended acceptance, and two reviewers recommended weak acceptance of the submission. Based on the above, I recommend acceptance and suggest that the authors address any remaining concerns raised by the reviewers in the final version.

**Audience:**

Yes

**Audience Explanation:**

Yes, some individuals in TMLR's audience would be interested in knowing the findings of this paper. Mainly researchers and practitioners who are work on interpretability methods for GNNs.

**Claims And Evidence:**

Yes

**Claims Explanation:**

The paper claims to study the problem of explaining node similarity in GNNs, which has not been addressed in prior work. To the best of my knowledge, this claim appears to be accurate. The paper also introduces criteria that explanations of similarity should satisfy, along with metrics to quantify these criteria after graph interventions. The three proposed criteria (actionable explanations, consistent explanations, sparse explanations) together with their corresponding metrics, are clearly presented in Section 4 in the manuscript. Finally, the paper includes experiments applying the proposed framework to mutual-information and gradient-based methods.